# Investigating Skin Microbial Community in Malignant Melanoma Lesions

**DOI:** 10.3390/microorganisms13050992

**Published:** 2025-04-25

**Authors:** Michele Properzi, Valentina Dimartino, Daniele Pietrucci, Carla Fontana, Claudia Rotondo, Luigi Lembo, Francesco Ricci, Francesca Scatozza, Giovanni Di Lella, Francesco Messina, Giovanni Chillemi, Barbara Bartolini, Antonio Facchiano

**Affiliations:** 1National Institute for Infectious Diseases “Lazzaro Spallanzani”—IRCCS, 00149 Rome, Italy; michele.properzi@inmi.it (M.P.); valentina.dimartino@inmi.it (V.D.); carla.fontana@inmi.it (C.F.); claudia.rotondo@inmi.it (C.R.); mss.francesco1984@gmail.com (F.M.); giovanni.chillemi@inmi.it (G.C.); 2Department for Innovation in Biological, Agro-Food and Forest Systems (DIBAF), University of Tuscia, 01100 Viterbo, Italy; 3Dermatologia Azienda Ospedaliera San Camillo Forlanini, 00152 Rome, Italy; luigilembo99@gmail.com; 4Istituto Dermopatico dell’Immacolata, IDI-IRCCS, FLMM, 00167 Rome, Italy; f.ricci@idi.it (F.R.); fscatozza@libero.it (F.S.); g.dilella@idi.it (G.D.L.); a.facchiano@idi.it (A.F.); 5Department of Experimental Medicine, University of Rome “Tor Vergata”, 00133 Rome, Italy

**Keywords:** skin microbiota, biomarker, melanoma, benign skin lesion

## Abstract

The skin microbiome is identified as one of the crucial factors in several pathological conditions, including its potential capacity in modulating cancer progression and response to treatment. A strong association of *Bacilli* and *Betaproteobacteria* classes and the *Bacteroidetes* phylum with melanoma is described in patients with cutaneous malignancies, while an imbalance of *S. epidermidis* and *S. aureus* is related to the progression of other skin cancers. In the present study, we characterized the microbial community in suspected lesions of 35 patients, classified, after histological analysis, as malignant melanoma lesions and benign non-melanoma lesions. Mirrored healthy skin were also included as negative control. No significant difference in alpha and beta diversity was observed when samples were categorized in four different groups (melanoma samples vs. contralateral healthy samples; melanoma samples vs. benign lesions; benign lesions vs. contralateral controls; melanoma controls vs. benign controls). The differential abundance analyses show that *Corynebacterium urealyticum* is more abundant in melanoma samples compared to their control, while *Roseomonas gilardii* is less abundant in melanoma. *Staphylococcus massiliensis*, *Bacillus coagulans*, *Paracoccus yeei*, *Corynebacterium jeikeium*, and *Corynebacterium pyruviciproducens* are present only in melanoma samples when compared with benign lesions.

## 1. Introduction

The skin microbiome is a barrier between the human body and the environment and plays a key role in preventing breaches by pathogenic microorganisms or hampering serious cutaneous degeneration. Disruption in the composition of the skin flora (diversity, quantity, composition, metabolism) is crucial in several pathological conditions. Certain microbial communities may lead to the onset or progression of various skin diseases such as acne, rosacea, psoriasis, or atopic dermatitis [1,2,3,4]. In atopic dermatitis, for example, *Staphylococcus aureus* and *S. epidermidis* predominate in severe inflammatory states, while after treatment there is an increase in *Streptococcus*, *Propionibacterium*, and *Corynebacterium* species [5].

In skin cancer, microbiota composition is identified as one of the crucial factors in modulating cancer progression and response to treatment [6], and the role of the microbiome in skin physiology and skin response to pathological conditions has been investigated [7,8,9,10] with some focus toward the modulation of response to immunotherapy and its influence in tolerability in patients with cutaneous malignancies [11]. Moreover, the composition of skin bacteria has been shown to influence melanoma. For instance, Zhu et al. [12] investigated the skin microbiota in 1656 skin samples and found strong associations of *Bacilli* and *Betaproteobacteria* classes and the *Bacteroidetes* phylum with skin melanoma. In addition, Giese and colleagues found that *S. aureus* from the skin microbiota is involved in melanoma invasion in a zebrafish model, likely via a lipase Sal2-mediated lipid synthesis [13], while an imbalance of *S. epidermidis* and *S. aureus* was found to be related to the progression of other skin cancers [14].

In the present study, we characterized the skin microbial community in malignant melanoma lesions and benign non-melanoma lesions to identify microbes associated with melanoma and explore a possible correlation whereby the abundance of specific microorganisms potentially lead to cancer.

## 2. Materials and Methods

### 2.1. Patient Enrollment

A total of 35 patients suspected of having melanoma were enrolled at the Department of Dermatology of IDI-IRCCS in Rome from 2016 to 2020, according to the protocol approved by the IDI Ethics Committee (ID n. 574-2017).

Before the surgical excision, two skin swabs were taken from each patient: one from the lesion site and another from a healthy area on the opposite side of the body. Topics application and antibiotics use was excluded at the patients enrollment. After the surgical excision, the patients were assigned to either the melanoma group (referred to as M-) or the benign lesion group (referred to as B-), based on the results of the histological analysis. Therefore, samples were classified as M-Samp (Melanoma lesion Swab Samples) or B-Samp (Benign lesion Swab Samples, all of which included nevi except one seborrheic keratosis) and the contralateral healthy body parts were labelled M-Ctrl (Melanoma Control) and B-Ctrl (Benign lesion Control), respectively.

Seventy skin swabs were meticulously collected and stored at −80 °C to ensure optimal preservation prior to DNA extraction.

### 2.2. Sequencing

After extraction with the DNA Mini Kit-Qiagen (Qiagen, Hilden, Germany), DNA was amplified with the Ion 16S Metagenomics kit (Ion Torrent™ Guilford, CT, USA) to target seven hypervariable regions of the 16S rDNA gene (the V2-4-8 and V3-6-7-9 regions). The libraries, prepared with the Ion Xpress Plus Fragment Library Kit (Ion Torrent™), were sequenced with the Ion GeneStudio™ S5 System (Ion Torrent™, ThermoFisher, Milan, Italy). To properly assess the sequencing data, control samples were also processed: two negative controls and two positive controls composed exclusively of *E. coli* sequences [15].

### 2.3. Bioinformatics Analysis

FastQC was used to evaluate the reads quality level (v0.11.9) and Trimmomatic was used to remove reads with low Phred score and adapters (v0.39). The sequencing data were analyzed using the QIIME2 pipeline (v2023.9.2). The dataset was loaded in the QIIME2 environment as .qza files, and reads were clustered in Amplicon Sequence Variants (ASVs) using the DADA2 (v2023.9.0) algorithm “qiime dada2 denoise-pyro”, a plug-in optimized for Ion Torrent reads. The representative sequence of each read was taxonomically assigned using the algorithm suite “qiime feature-classifier classify-sklearn” in QIIME2 (v 2023.9.0), and the SILVA database as reference (Silva 138) [16]. The files obtained using the QIIME2 pipeline were then used as input for statistical analyses in R (v4.3.2) using the qiime2R package (v0.99.6). Statistical analyses were performed using the following packages: vegan (v2.6-4), phyloseq (v1.46.0), DESeq2 (v1.42.1), rstatix (v0.7.2), and pwr (v1.3-0). Firstly, low-abundance ASVs were removed from the analysis [17]; subsequently, the samples were normalized using the rarefaction approach [18] and the taxa were analyzed at the level of species.

The alpha diversity analysis was obtained through the vegan R package and was carried out by applying four different metrics: Observed Species, Shannon index, Simpson index, and Fisher index. Differences in alpha diversity across sample categories were assessed using the Kruskal–Wallis rank-sum test, followed by Dunn’s post hoc test. The *p*-values of Dunn’s post-hoc tests were corrected using a False Discovery Rate correction. The power and the effect size of the test were computed for all the alpha-diversity analyses, to take into account the Type II error [19]. The beta diversity was evaluated using four different metrics: Bray–Curtis dissimilarity, Canberra Distance, and Weighted and Unweighted Unifrac distances. Differences across sample categories in terms of beta diversity were evaluated using a PERMANOVA test, with 9999 permutations. The differential abundance analysis was carried out using the Wald test through the DESeq2 package, as previously reported for similar datasets [20]. The graphical representation was achieved through EnhancedVolcano (v1.20.0), imposing LogFC |1.5| and False Discovery Rate (FDR) 0.05 to highlight only statistically significant species.

## 3. Results

### 3.1. Samples Selection

A total of 70 samples were sequenced and then trimmed, obtaining a total of 24.5 million reads, with a mean of 350,032 reads per sample. After performing the clustering analysis using the QIIME2 pipeline (Appendix A) and removing the low-frequency ASVs, the samples were characterized by 8460 reads per sample (Figure 1A). Two negative controls and two *Escherichia coli* samples were sequenced together with the 70 clinical samples. The analysis of negative control samples identified species that, present in low amounts, were considered contaminants in the melanoma samples (i.e., *Amia calva*, *Brevibacterium pityocampae*, *Paracoccus marinus*, *Ammopiptanthus mongolicus*, *Lactococcus piscium*) and were removed from the analyses. As for the *Escherichia coli* control samples, the results are consistent with expectations, as on average, 50.8% of reads are associated with the order *Enterobacterales*, 41.8% with the family *Enterobacteriaceae*, and 30.77% with the genus *Escherichia*.

Subsequently, the samples reads were normalized using the rarefaction method. We plotted the rarefaction curve on the 70 samples (Figure 1B) and imposed a threshold on the number of reads per sample of 3.000. This procedure discarded 20 samples, leaving 50 samples for analysis as summarized in Table 1. A total of 9 phyla, 13 classes, 22 orders, 30 families, 39 genera, and 70 species were identified, on which a summary phylogenetic tree was built (Figure 1C).

### 3.2. Alpha and Beta Diversity Analysis

Alpha and beta diversity and differential abundance analyses were performed at the species taxonomic level. Samples were categorized according to their status in the following categories: melanoma (M-Samp, 17 samples), benign lesion (B-Samp, 7 samples) and the respective healthy categories (M-Ctrl, 19 samples and B-Ctrl, 7 samples). A comparison of alpha diversity between the four groups of samples (Figure 2A) showed that no significant differences in microbial community were found for the Shannon index (Kruskall–Wallis *p*-value = 0.5531, all Dunn post hoc test were greater than 0.05). Similar results were obtained using three other alpha diversity metrics: Number of Observed Species, Simpson, and Fisher. These results were evaluated in term of effect size [19] and statistical power. The effect size estimates indicated a low effect size (eta-squared ≤ 0.1) and, additionally, the power of our study was estimated to be only 25%, indicating that the study may not have had sufficient power to detect small or medium effect sizes in terms of alpha diversity. Regarding beta diversity, no significant differences were identified among the four groups. Figure 2B shows the results of the Principal Coordinate Analysis (PCoA) performed using the Bray–Curtis metric, which yielded a non-significant result (PERMANOVA *p*-value = 0.3864). Similar results were obtained with other beta diversity metrics, such as Canberra distance, Weighted UniFrac, and Unweighted UniFrac. The results of the beta diversity may be influenced by the small number of samples, as previously described for the alpha diversity.

### 3.3. Differential Abundance Analysis

Considering the presence of four sample classes, the following four comparisons were performed to identify bacteria with differential abundance:Melanoma samples compared to paired contralateral healthy samples (M-Samp vs. M-Ctrl) (paired analysis);Melanoma samples compared to benign lesions (M-Samp vs. B-Samp) (uncoupled analysis);Benign lesions compared to paired contralateral controls of benign lesions (B-Samp vs. B-Ctrl); (paired analysis);Melanoma controls compared to controls of benign lesions (M-Ctrl vs. B-Ctrl) (unpaired analysis).

Due to the limited sample size, we were unable to adjust for demographic and clinical covariates. This limitation may have contributed to potential confounding effects for each comparison. The evaluation across several comparisons allowed us to potentially discriminate bacteria that may vary across individuals (due to covariates) and not correlate with the melanoma. In detail, Comparison 2 and Comparison 4 represent bacteria that may vary across individuals. Comparison 1 (M-Samp vs. M-Ctrl) identifies bacteria that are differentially abundant in melanoma tissues compared to contralateral healthy tissues within the same individual, allowing for the selection of potential biomarkers. Four bacteria were found to show statistically significant different abundance in this comparison: *Roseomonas gilardii*, *Lactobacillus hominis*, *Corynebacterium striatum*, and *Corynebacterium urealyticum*. *Roseomonas gilardi* and *Corynebacterium striatum* were present in lower abundance in melanoma samples, while *L. hominis* and *C. urealyticum* were more abundant in melanoma lesions.

Comparison 3 (B-Samp vs. B-Ctrl) highlights potential microbial differences in benign lesions compared to healthy tissues within the same individual and serves as a reference for identifying bacteria variability in different skin areas of healthy individuals. In this case, four bacteria were found with a statistically significant abundance: *Corynebacterium macginleyi*, *Campylobacter ureolyticus*, *Corynebacterium simulans*, and *Peptoniphilus harei*. *C. simulans* was less present in lesions, while *C. macginleyi*, *C. ureolyticus*, and *P. harei* were more abundant.

Finally, Comparisons 2 and 4 (M-Samp vs. B-Samp and M-Ctrl vs. B-Ctrl) perform unpaired analyses and aim to identify microorganisms variability among individuals. These comparisons identified statistically significant different abundance of 15 and 12 bacteria, respectively. The high number of significant bacteria in these latter comparisons may result from the absence of paired comparisons, leading to an increased number of identified species due to individual variability.

A heatmap was generated to illustrate all comparisons, displaying the fold changes and indicating whether each species is more or less abundant in a given comparison (Figure 2C).

Several species were identified with statistically significant differential abundance in more than one comparison, including the following:*Lactobacillus hominis*: strongly increased in the M-Samp vs. M-Ctrl and strongly decreased in M-Ctrl vs. B-Ctrl;*Corynebacterium striatum*: strongly decreased in M-Samp vs. M-Ctrl and in M-Samp vs. B-Samp and strongly increased in M-Ctrl vs. B-Ctrl;*Campylobacter ureolyticus* and *Peptoniphilus harei*: both strongly reduced in M-Samp vs. B-Samp and strongly increased in B-Samp vs. B-Ctrl;*Corynebacterium macginleyi*: both strongly increased in B-Samp vs. B-Ctrl and M-Ctrl vs. B-Ctrl and strongly decreased in M-Samp vs. B-Samp;*Corynebacterium glucuronolyticum, Staphylococcus pettenkoferi, Acinetobacter ursingii, Streptococcus dysgalactiae, Corynebacterium kroppenstedtii, Anaerococcus prevotii:* differently present in M-Samp vs. B-Samp and M-Ctrl vs. B-Ctrl.

On the other hand, the species identified with statistically significant differential abundance only in a single comparison are as follows:*Corynebacterium urealyticum*, *Roseomonas gilardii*: present only in the M-Samp vs. M-Ctrl comparison;*Staphylococcus massiliensis, Bacillus coagulans, Paracoccus yeei, Corynebacterium jeikeium, Corynebacterium pyruviciproducens*: present only in M-Samp vs. B-Samp comparison;*Corynebacterium simulans*: present only in B-Samp vs. B-Ctrl;*Streptococcus anginosus, Brevibacterium casei, Prevotella timonensis*: present only in M-Ctrl vs. B-Ctrl.

Two species were identified only comparing the melanoma lesions and their healthy contralateral part (M-Samp vs. M-Ctrl): *R. gilardii* and *C. urealyticum*. The *C. simulans* was identified only comparing the benign lesions to the corresponding contralateral samples (B-Samp vs. B-Ctrl).

Overall, contrary to the alpha and beta diversity analyses which did not yield significant results, the differential abundance analysis allowed us to locate *C. urealyticum* and *L. hominis* as more abundant in melanoma samples compared to the respective controls, and *R. gilardii* as less abundant in melanoma samples compared to the respective controls, while *S. massiliensis*, *B. coagulans*, *P. yeei*, *C. jeikeium*, and *C. pyruviciproducens* are more abundant in melanoma samples compared to samples from benign lesions.

## 4. Discussion

Oncological therapies are rapidly evolving, testing increasingly targeted and innovative approaches [21,22]. Recently, the investigation of the microbiota has also emerged as a promising area of research [20].

The focus of this study was to compare the characteristics of the skin microbiota under different conditions. Starting with skin swabs taken at different body sites, we analyzed how the cutaneous microbiota changes, focusing on possible genera and species related to the melanoma environment.

The alpha and beta diversity analyses did not allow us to draw significant information from the data; overall, it is possible that the limited number of samples may have negatively affected the visualization of these metrics.

However, several bacterial genera emerged with significant differential abundance, and the considerations developed for those related to the context of melanoma are set out below.

*Corynebacterium* species were found to be altered in Comparison 1 (M-Samp vs. M-Ctrl) but also in other comparisons. Specifically, *C. urealyticum* was more abundant in melanoma samples and was found only in Comparison 1 (M-Samp vs. M-Ctrl). *C. striatum* was identified in Comparison 1, but also in comparisons between different categories of samples. *C. macginleyi*, *glucuronolyticum*, and *kroppenstedtii* were also identified in other comparisons, but not in melanoma samples.

*Corynebacterium* is a genus known to be related to melanoma, in particular with stage III and IV melanomas. Patients positive for the *Corynebacterium* genus, in fact, showed a higher number of cells positive for interleukin 17 (IL-17) [6]. It is noticeable that IL-17 can promote the growth of melanoma by up-regulation of IL-6 and the signal transducer and activator of transcription 3 [23], demonstrating the possible involvement of *Corynebacterium* in the appearance of melanoma [6].

However, it should be considered that this genus includes species that act as simple commensals of the skin, like *C. urealyticum* [24]. Our data, therefore, indicate that further investigations are needed to clarify the role of *Corynebacterium* species like *Corynebacterium urealyticum* in melanoma dynamics.

*S. massiliensis* shows significant differential abundance on melanoma samples when compared with benign lesions (M-Samp vs. B-Samp); however, it appears to be a skin commensal [25], making it possible to hypothesize a role in melanoma setup for this commensal microorganism.

We also observed that *R. gilardii* was significantly less abundant in melanoma samples. Among the 15 identified species within this genus, only *R. gilardii* and *R. mucosa* are known to have potential pathogenic effects on humans, with *R. gilardii* being a rare cause of bacteremia in immunocompromised individuals [26]. However, other studies indicate that *R. gilardii* can be found in the normal skin microbiota [27], suggesting that further investigations are necessary for this bacterium as well.

Of significant interest, potential bacterial biomarkers for melanoma were identified: *R. gilardii* and *C. urealyticum*. Of these, considering only statistically significant data, *R. gilardii* is more rarefied only on melanoma samples, while *C. urealyticum* is more abundant only on melanoma samples.

It is however true that, as already mentioned, these two species fall within the field of commensal skin bacteria, so further in-depth and targeted analyses would be necessary to provide greater validity to this assumption for these two bacteria.

Immune checkpoint inhibitor (ICI) therapy significantly improved melanoma management [28]. The patient’s response to ICI is not always effective, and taking probiotics can have a beneficial effect from this point of view, although this is a subject of discussion [29]. *L. hominis*, a microaerophilic facultative anaerobe bacterium [30], emerged with greater abundance in melanoma samples (M-Samp vs. M-Ctrl), but was also identified with lower abundance in M-Ctrl samples (M-Ctrl vs. B-Ctrl). It is worth noting that another species of the genus *Lactobacillus*, *Lactobacillus reuteri*, is able to colonize preclinical melanomas to support ICI therapy. *L. reuteri* promotes the activation of anti-tumor CD8+ IFNγ+ T cells by releasing indole-3-aldehyde (I3A). High serum levels of I3A were indeed found in patients who responded well to ICI therapy. These data suggest that a dietary intervention with probiotics could have a beneficial effect on patients undergoing ICI [29,31]. This suggests that, in addition to *L. reuteri*, other species within the *Lactobacillus* genus, like *L. hominis*, may also exhibit similar anti-melanoma effects.

The species present in a higher proportion in controls of melanoma patients than in controls of benign lesions (M-ctrl vs. B-Ctrl) may suggest an initial favorable condition for a pathological skin event; for example, the presence of *Streptococcus dysgalactiae* and *S. anginosus*, belonging to beta-hemolytic group C and G streptococci (GCGS), may suggest a predisposition to an inflammatory state [32,33].

Generally, histological examination reveals an adjacent nevus remnant in 30% of cutaneous melanoma cases, while the remaining 70% are de novo melanomas. However, the dynamics underlying melanoma development from a pre-existing melanocytic nevus versus its de novo appearance remain a topic of debate [34]. Considering this, attention was also given to species showing a significant differential abundance in benign lesions that appeared suspect enough to suggest being excised. *Peptoniphilus* is a genus involved in prostate, endometrial, breast, bladder, oral, and cervical cancer, in which it appears to have a greater abundance [35]. We found an increased abundance of *P. harei* in benign lesions suspected to be pathological and requiring excision when compared to the healthy contralateral samples (B-Samp vs. B-Ctrl), but, at the same time, it is less abundant in melanoma samples when compared to benign lesions (M-Samp vs. B-Samp). This leads to questioning its involvement with this type of tumor. *C. urealyticus* is known to be more abundant together with other anaerobes on skin subject to hidradenitis suppurativa (also known as acne inversa or Verneuil’s disease, a chronic inflammatory condition of the skin) [36], and also this bacterium was found to be more abundant in benign lesions when compared to the healthy contralateral samples (B-Samp vs. B-Ctrl) but less abundant in melanoma samples when compared to benign lesions (M-Samp vs. B-Samp), making it difficult to bring this bacterium closer to the context of melanoma.

## 5. Conclusions

The bacterial composition in melanoma shows high similarity not only between samples from the same patient, but also between samples from different patients, thus allowing us to consider that there are bacterial species characteristic of melanoma [37], and therefore there must be factors that act to “shape” the composition of the microbiota towards this recurring composition, inspiring interest in being able to characterize the relative microbiota. Our study suffers from some limitations, particularly the imbalanced numerosity of the melanoma group and benign group, which may have influenced the alpha and beta diversity results. Nevertheless, the robustness of the other statistical analyses warrants the statistical relevance of the collected results. This limitation is a common challenge in studies with small cohorts, but we believe that our preliminary results are important for future studies on larger cohorts. Increasing our understanding of the relationship between melanoma and the microbiota could enhance our diagnostic capabilities for this tumor. By incorporating microbiota analysis into existing diagnostic methods, we may be able to intervene directly at the skin level. This could involve altering the composition of the skin microbiota to reduce the presence of species that promote tumor growth, while encouraging the proliferation of species that may help combat it.

## Figures and Tables

**Figure 1 microorganisms-13-00992-f001:**
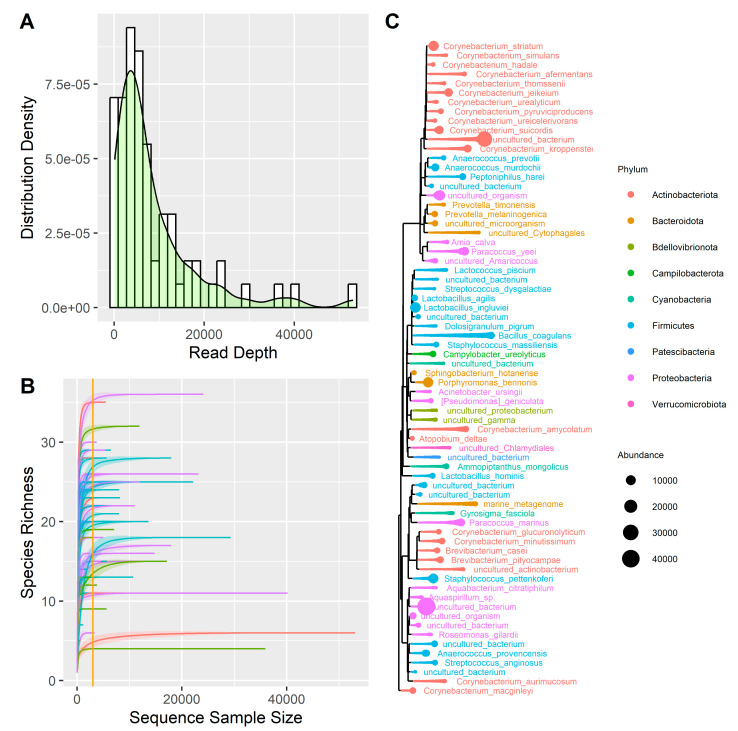
(**A**) Distribution of reads depth. (**B**) Rarefaction curve; the curves represent the following: purple = M-Samp, blue = M-Ctrl, green = B-Samp, red = B-Ctrl. (**C**) Phylogenetic tree representing all the species found in the samples, distinguished by phylum and abundance.

**Figure 2 microorganisms-13-00992-f002:**
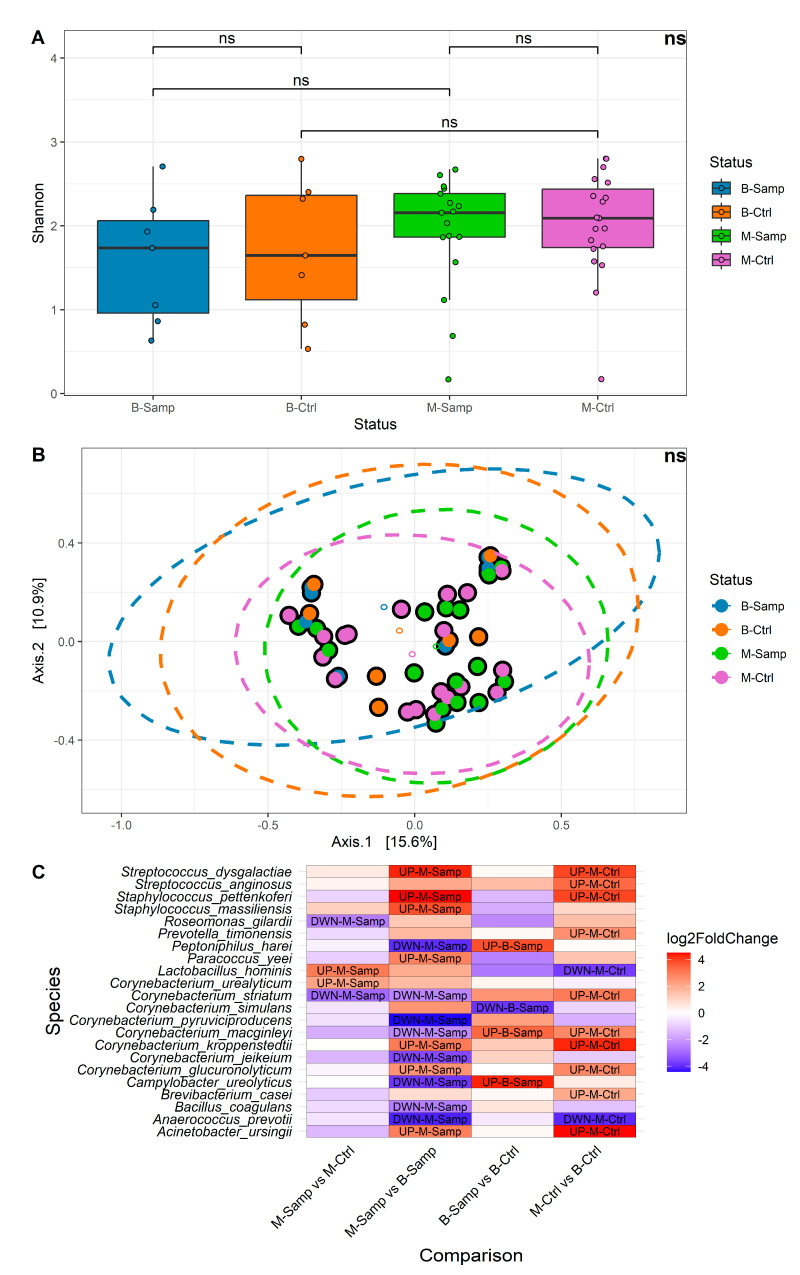
(**A**) Boxplots of alpha diversity by Shannon metric. (**B**) Bray–Curtis graph for beta diversity. (**C**) Heatmap showing the abundance levels of each species with significant differential abundance for the four comparisons evaluated. In the heatmap, fold change indicates differential abundance of the species. Specifically: (1) In the M-Samp vs. M-Ctrl column, positive fold change indicates higher abundance in M-Samp; (2) In the M-Samp vs. B-Samp column, positive fold change indicates higher abundance in M-Samp; (3) In the B-Samp vs. B-Ctrl column, positive fold change indicates higher abundance in B-Samp; (4) In the M-Ctrl vs. B-Ctrl column, positive fold change indicates higher abundance in M-Ctrl. “ns” means “not significant”.

**Table 1 microorganisms-13-00992-t001:** Summary of data relating to the 50 samples analyzed.

	Cohort	Swabs		Controls
	Disease	M-Samp n = 17	B-Samp n = 7	M-Ctrl, B-Ctrl n = 19 + 7
	Mean Age	61	46	56
	Gender (F/M) *	5/10	4/2	9/14
**Body Skin Composition**	Sebaceous	8	3	11
	Moist	5	3	9
	Dry	4	1	6
**Photo-exposure**	Yes	14	6	22
	No	3	1	4
**Ulceration**	Yes	2	-	-
	No	15	-	-
	NA	-	7	26
**Tumor stage**	T1	14	-	-
	T2	1	-	-
	T3	1	-	-
	T4	1	-	-
	NA	-	7	26
**Intratumoral lymphocyte infiltrate**	Yes	12	-	-
	No	5	-	-
	NA	-	7	26
	Mean Breslow thickness	1 mm	-	-

M-Samp: melanoma sample; B-Samp: benign sample; M-Ctrl: melanoma sample control; B-Ctrl: benign lesion sample control. * not included: 6 patients with unknown gender.

## Data Availability

The original data presented in the study are openly available in NCBI—SRA at BioProject ID PRJNA1255084.

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
