# Peer review of "Investigating Skin Microbial Community in Malignant Melanoma Lesions"

_microorganisms, 2025, doi:10.3390/microorganisms13050992_

Round 1
Reviewer 1 Report
Comments and Suggestions for Authors
- The melanoma group (n=17) and benign group (n=7) are imbalanced, reducing statistical power. Post-rarefaction exclusion of 29% of samples (20/70) risks biasing results, particularly if excluded samples had distinct microbial profiles.
- DESeq2, designed for RNA-seq, was applied to compositional microbiome data without addressing compositionality (e.g., via centered log-ratio transformation). This may inflate false positives in differential abundance analysis.
- To enhance the background discussion on antitumor applications, consider referencing the following studies: 10.1038/s41563-019-0503-4 and 10.1021/acsbiomaterials.5b00346. Additionally, highlighting the unique contributions of this research compared to previously published work in the field would strengthen the paper.
- Non-significant alpha/beta diversity results (Figure 2A-B) are presented without acknowledging potential Type II errors due to small sample sizes. Overlapping Shannon index distributions in Figure 2A visually suggest limited power.
- The heatmap uses strings ("UP/DOWN") instead of color gradients to represent abundance, hindering intuitive interpretation. A traditional heatmap with hierarchical clustering would better illustrate microbial patterns.
- Demographic variables (e.g., age, gender, skin type) and clinical factors (e.g., Breslow thickness, ulceration) in Table 1 were not adjusted for in analyses, potentially confounding microbial associations.
- Figure 1C: Phylogenetic tree lacks clear labeling (e.g., phylum/node annotations) and a legend, limiting interpretability. Figure 1B: Rarefaction curves (purple, blue, green, red) are inadequately described in the caption, leaving readers to infer group-color mappings.
Author Response
Manuscript ID: microorganisms-3540613
Response to Reviewer
We thank Reviewer for the valuable evaluation of the manuscript. We have properly addressed all comments and believe that the revised version has improved in both clarity and robustness.
Please find below a point-by-point reply.
Point-by-point reply to Reviewer
Comment 1
The melanoma group (n=17) and benign group (n=7) are imbalanced, reducing statistical power. Post-rarefaction exclusion of 29% of samples (20/70) risks biasing results, particularly if excluded samples had distinct microbial profiles.
Reply 1
We thank the Reviewer for this comment. We agree that the two groups are imbalanced; this is due to the study design, since the nature of lesion was only “suspected” at the enrolment and then histologically confirmed afterword. This is clearly recognized as a potential limitation of the study and reported in the Conclusions paragraph, at lines 366-369. At the same time, we are aware that the rarefaction may reduce the number of samples. However, recently, the rarefaction has been reported as a good method when differences regarding the sequencing depth across samples is present (doi:10.1128/msphere.00354-23). We cited that study and used that approach in order to reduce the number of false positives in the differential abundance analysis.
To comply with the reviewer’s comment, we introduced an additional sentence in the conclusion as further caution sentence (lines 366-371).
Comment 2
DESeq2, designed for RNA-seq, was applied to compositional microbiome data without addressing compositionality (e.g., via centered log-ratio transformation). This may inflate false positives in differential abundance analysis.
Reply 2
We acknowledge the Reviewer’s concern regarding the application of DESeq2 to compositional microbiome data. While it is true that DESeq2 was originally developed for RNA-seq data, it has also been widely applied in microbiome studies, including work published recently (doi:10.1001/jamadermatol.2023.2955). Moreover, the DESeq2 framework does not explicitly require data transformation to account for compositionality, and its use has become relatively standard in differential abundance analyses of metagenomic data. We appreciate the reviewer’s suggestion and will explicitly cite the aforementioned study in the revised manuscript to support our methodological choice (line 109).
Comment 3
To enhance the background discussion on antitumor applications, consider referencing the following studies: 10.1038/s41563-019-0503-4 and 10.1021/acsbiomaterials.5b00346. Additionally, highlighting the unique contributions of this research compared to previously published work in the field would strengthen the paper.
Reply 3
We thank the Reviewer for the suggestion, possible applications for cancer treatments are now mentioned in the revised version of the manuscript (lines 278-280).
Comment 4
Non-significant alpha/beta diversity results (Figure 2A-B) are presented without acknowledging potential Type II errors due to small sample sizes. Overlapping Shannon index distributions in Figure 2A visually suggest limited power.
Reply 4
We thank the Reviewer for this important point. We agree that the non-significant results observed for alpha/beta diversity (Figure 2A-B) may be due to potential Type II errors arising from the small sample size. To address this concern, we conducted a post hoc power analysis using the effect size (https://doi.org/10.1093/bib/bbab471) and power test. The packages used have been added to the manuscript (lines 94-95). The latter one was executed using the pwr.anova.test function in R software, by using the medium number of samples for each category, considering the unbalance sampling across groups. Based on the observed variability (e.g., Shannon index) we found that our study's power is insufficient to detect small or medium effects. We have now included this limitation in the revised version of the manuscript, in the materials and methods and the results sections (lines 102-104 and lines 158-162). Regarding the calculation of Type II errors for the PERMANOVA test used in the analysis of beta-diversity, we are not currently aware of a standardized method to directly calculate Type II errors for this specific test. However, we acknowledge that the small sample size may have reduced the statistical power to detect significant differences between groups (lines 167-168).
Comment 5
The heatmap uses strings ("UP/DOWN") instead of color gradients to represent abundance, hindering intuitive interpretation. A traditional heatmap with hierarchical clustering would better illustrate microbial patterns.
Reply 5
We thank the Reviewer for this helpful suggestion. We have modified the figure accordingly and added further details to the figure caption to provide more information regarding the fold change for each comparison, since the “UP/DOWN” string was used in order to better specify the information regarding the fold change (lines 197-202).
Comment 6
Demographic variables (e.g., age, gender, skin type) and clinical factors (e.g., Breslow thickness, ulceration) in Table 1 were not adjusted for in analyses, potentially confounding microbial associations.
Reply 6
We thank the Reviewer for raising this important point. We fully acknowledge that adjusting for demographic and clinical covariates (such as age, gender, skin type, Breslow thickness, and ulceration) is considered best practice in microbiome studies. Indeed, such adjustments are performed in other recent studies, including the one we cited earlier (doi: 10.1001/jamadermatol.2023.2955). However, due to the limited sample size in our study, it was not statistically feasible to include multiple covariates in the differential abundance analysis. The inclusion of several covariates substantially reduces the degrees of freedom, and in our case, resulted in model convergence issues or failure of the statistical packages to perform the analysis. This limitation is a common challenge in studies with small cohorts and it will certainly be useful to conduct a future study with a larger cohort of patients.
Given these limitations, we will explicitly acknowledge this point in the revised manuscript and discuss the potential confounding effects as part of the study’s limitations (lines 215-220).
Comment 7
Figure 1C: Phylogenetic tree lacks clear labeling (e.g., phylum/node annotations) and a legend, limiting interpretability. Figure 1B: Rarefaction curves (purple, blue, green, red) are inadequately described in the caption, leaving readers to infer group-color mappings.
Reply 7
The image has been modified according to the request.
Reviewer 2 Report
Comments and Suggestions for Authors
REVIEW:
In this piece of research, the Authors, M. Properzi and collaborators, addressed the differential composition of the microbial community in melanoma lesions. To characterize the microbiome of melanomas the Authors used an NGS-based approach (Ion Torrent methodology) to interrogate microbial 16S ribosomal DNA. Appropriate sequencing controls have been introduced, and the pipelines for data obtention, curation and analysis look robust. The microbiome of melanomas has been systematically compared with that of contralateral healthy skin in paired analyses, and with the microbiomes of benign lesions and healthy skin samples in uncoupled (and unpaired) analyses. Alpha and beta diversities of microbial populations, within and between samples, were each assessed using different, established metrics. Although no significant differences in alpha or beta diversities between melanomas and benign lesions and normal skin could be established, differential abundance analyses revealed imbalances in melanoma lesions regarding some species, with some being higher (or lower) abundance in melanoma samples.
The topic is relevant, with current evidence supporting a role for the microbiome in the host response against melanoma tumours and in the modulation of the response to immunotherapies.
The article is descriptive but reads well, and experiments look well conducted. However, there are some caveats with this research, discriminated below, which the Reviewer believes can be addressed by the Authors. These criticisms are solely intended at improving the quality of this research.
Comments:
- The number of melanoma cases is low (n=17), yet within an acceptable range. The melanomas should be early stage (mostly T1, low average Breslow index), but two are ulcerated; do these have significant differences in the microbiome (outliers)?? If so, data regarding these lesions could be presented also separately, eg in Suppl. Data.
- To our understanding, the sampling was superficial (swabbing). This shall be mentioned and discussed in Discussion section, since intra-tumoral (including intracellular) microbiomes are also currently being assessed in solid tumours.
- Which cautionary measures were patients instructed to follow before sampling? Skin washing and application of topics (common among patients attending dermatologic clinics and wards) may deeply alter the superficial microbiome.
- Subsections in Results section end up abruptly; some wrap-up/sum-up of the relevant findings would add to clarity of the main messages.
- In Figure 2 (C), it is intriguing the difference in microbiota between the normal (control) skin of melanoma patients and the normal (control) skin of patients bearing benign lesions. The Authors should comment on this.
- What is the nature of the benign lesions? Pigmented lesions (eg, lentigos, nevi) that might be confused with melanoma? This should be mentioned in Materials and Methods and Results.
- Minor corrections suggested: a) the sentence between lines 304-310 is too long; it could be fragmented; b) in line 275…”…IL17 can promote the development of melanoma” … should be modified to, for example, ” IL17 can promote the growth of melanoma..”. As it stands it appears that IL17 in involved in carcinogenesis, for which evidence is missing in melanoma, to the best of our knowledge.
Author Response
Manuscript ID: microorganisms-3540613
Response to Reviewer
We thank Reviewer for the valuable evaluation of the manuscript. We have properly addressed all comments and believe that the revised version has improved in both clarity and robustness.
Please find below a point-by-point reply.
Point-by-point reply to Reviewer
Comment 1
The number of melanoma cases is low (n=17), yet within an acceptable range. The melanomas should be early stage (mostly T1, low average Breslow index), but two are ulcerated; do these have significant differences in the microbiome (outliers)?? If so, data regarding these lesions could be presented also separately, eg in Suppl. Data.
Reply 1
We thank the Reviewer for the comment. Patients enrolment has been closed and unfortunately we cannot enroll additional patients. To comply with this issue, we added statements in the revised version of the manuscript to highlight such limitation; however, we also mention that any effort has been spent to warrant the statistical significance of the results (lines 366-369). Regarding the samples classified as ulcerated, we conducted a beta-diversity analysis and found that they did not behave differently from the other samples. Therefore, we decided not to include this information in the supplementary material.
Comment 2
To our understanding, the sampling was superficial (swabbing). This shall be mentioned and discussed in Discussion section, since intra-tumoral (including intracellular) microbiomes are also currently being assessed in solid tumours.
Reply 2
We thank the Reviewer for this clarification. We confirm that sampling was carried out by swabbing. As requested, this aspect has been highlighted in the manuscript (lines 281-284).
Comment 3
Which cautionary measures were patients instructed to follow before sampling? Skin washing and application of topics (common among patients attending dermatologic clinics and wards) may deeply alter the superficial microbiome.
Reply 3
Thanks for this clarification. Topics application and antibiotics use was excluded at the patients enrolment (lines 65-66).
Comment 4
Subsections in Results section end up abruptly; some wrap-up/sum-up of the relevant findings would add to clarity of the main messages.
Reply 4
We thank the Reviewer. As requested, we have added a short summary paragraph to name the species that were found to be of greatest interest by the differential abundance analysis (lines 271-276).
Comment 5
In Figure 2 (C), it is intriguing the difference in microbiota between the normal (control) skin of melanoma patients and the normal (control) skin of patients bearing benign lesions. The Authors should comment on this.
Reply 5
We thank the Reviewer for pointing this out. We have added a discussion of this aspect with a possible hypothesis explaining this result (lines 337-341).
Comment 6
What is the nature of the benign lesions? Pigmented lesions (eg, lentigos, nevi) that might be confused with melanoma? This should be mentioned in Materials and Methods and Results.
Reply 6
Thank you for the clarification. In the revised version of the manuscript we completed the missing data. All the benign lesions are nevi, except one seborrheic keratosis (lines 70-71).
Comment 7
Minor corrections suggested: a) the sentence between lines 304-310 is too long; it could be fragmented; b) in line 275…”…IL17 can promote the development of melanoma” … should be modified to, for example, ” IL17 can promote the growth of melanoma..”. As it stands it appears that IL17 in involved in carcinogenesis, for which evidence is missing in melanoma, to the best of our knowledge.
Reply 7
- According to the requests, we rephrased the sentence in order to be more clear (lines 329-334):
“It is worth noting that another species of the genus Lactobacillus, Lactobacillus reuteri, is able to colonize preclinical melanomas to support ICI therapy. L. reuteri promotes the activation of anti-tumour CD8+ IFNγ+ T cells by releasing indole-3-aldehyde (I3A). High serum levels of I3A were indeed found in patients who responded well to ICI therapy. These data suggest that a dietary intervention with probiotics could have a beneficial effect on patients undergoing ICI”.
- Thanks for the suggestions provided, we have modified the text accordingly (lines 299-302)